# Influence of Land Use on Avian Diversity in North African Urban Environments

**Hani Amir Aouissi** [1,2], **Alexandru-Ionuţ Petrişor** [3,4,*], **Mostefa Ababsa** [1], **Maria Boştenaru-Dan** [5], **Mahmoud Tourki** [6] and **Zihad Bouslama** [2,7]

1. Scientific and Technical Research Center on Arid Regions (CRSTRA), Biskra 07000, Algeria; aouissi.amir@gmail.com (H.A.A.); mustapha.ababsa@gmail.com (M.A.)
2. Faculty of Sciences, Department of Biology, Badji-Mokhtar Annaba University, Annaba 23000, Algeria; ihadb@yahoo.fr
3. Doctoral School of Urban Planning, Ion Mincu University of Architecture and Urbanism, 010014 Bucharest, Romania
4. National Institute for Research and Development in Tourism, 50741 Bucharest, Romania
5. Department for the Management of Research, Ion Mincu University of Architecture and Urbanism, 010014 Bucharest, Romania; Maria.Bostenaru-Dan@alumni.uni-karlsruhe.de
6. Soil and Sustainable Development Laboratory, Faculty of Earth Sciences, Badji-Mokhtar Annaba University, Annaba 23000, Algeria; mahmoud.tourki23@gmail.com
7. Environmental Research Center (CRE), Badji-Mokhtar Annaba University, Annaba 23000, Algeria
* Correspondence: alexandru.petrisor@uauim.ro; Tel.: +40-213-077-191

**Abstract:** Land cover and use changes are important to study for their impact on ecosystem services and ultimately on sustainability. In urban environments, a particularly important research question addresses the relationship between urbanization-related changes and biodiversity, subject to controversies in the literature. Birds are an important ecological group, and useful for answering this question. The present study builds upon the hypothesis according to which avian diversity decreases with urbanization. In order to answer it, a sample of 4245 observations from 650 sites in Annaba, Algeria, obtained through the point abundance index method, were investigated by computing Shannon-Wiener's diversity index and the species richness, mapping them, and analyzing the results statistically. The findings confirm the study hypothesis and are relevant for planning, as they stress the role of urban green spaces as biodiversity hotspots, and plead for the need of connecting them. From a planning perspective, the results emphasize the need for interconnecting the green infrastructure through avian corridors. Moreover, the results fill in an important lack of data on the biodiversity of the region, and are relevant for other similar Mediterranean areas. Future studies could use the findings to compare with data from other countries and continents.

**Keywords:** urbanization; diversity indices; Annaba; geo-statistical approaches; urban ecology; bird abundance

## 1. Introduction

The crucial importance of studying land cover and use changes (LCUC) results from their being part of the "global changes" [1], constituting a major component [2]. They transform natural into man-dominated systems [3], threatening biodiversity [4], affecting the quality of water, land and air resources, ecosystem processes, functions, services, equilibrium, and resilience, and making an important impact on the climate system [5–9]. For these reasons, studying LCUC makes an important contribution to the sustainable development debate [6,10]. LCUC are more intense when associated with land fragmentation [11], and constitute an important threat for coastal areas [12].

Some authors consider that modernization [13] particularly migration to urban centers [14], is an important driver of LCUC. A supporting argument is the fact that an important LCUC, urbanization, is now considered a major driving force of biodiversity loss

and biological homogenization [15], generating a disproportionate share of environmental impacts compared to the total area affected by it [16]. The effects include habitat fragmentation [16–20], which in its turn influences species and biogeochemical cycles [18]. Urban sprawl is also a main threat to non-urbanized areas [16,21]. The Habitat II conference introduced the concept of "ecological footprint" to measure how much the new constructions, including infrastructure, affect the environment [22].

Over time, urban ecologists studied the ecology in the city, ecology of the city, and urban sustainability [23]; researches have classified the nature of cities in four categories: remains of natural systems, extensions of natural systems, landscaped or managed areas, and spontaneous, invasive, or ruderal species [24,25]. As a consequence of fragmentation, urban ecosystems are characterized by a built matrix embedding natural corridors and small, fragmented patches, and processes including succession and invasion [26–29].

While the overall influence of fragmentation on the urban ecosystem has been studied from a plethora of perspectives, its direct connection with biodiversity is still debated in the literature. Some authors associate fragmentation with a low biodiversity of small isolated patches [30,31], especially because species within these patches are more exposed to anthropogenic impacts [31,32] and have a reduced areal [33]. As a consequence, the size of patches is considered a good predictor of species richness [31,34]. Other authors found out that urban sprawl reduces species richness, but the abundances of some species might peak due to edge effects [27,34]. Moreover, even rare and endangered species can be preserved in urbanized habitats [26,35] when there is enough connectivity to provide corridors for certain species [34]. On an intermediate position, some authors consider that urban biodiversity depends on the spatial structure (understood as size of habitats, and distance between them) [19]. In addition, the relationship of different species with humans (hemerophobic, hemerodiaphoric, or hemerophile) plays an important role [36]. Roden [37] approaches the same issue as Mollov [36], differentiating between various species in the city and also highlighting the need for avian biodiversity. More specifically, for birds it is not so important to have continuous green spaces (assured by connecting green areas and through the presence of trees alongside of roads) but, because of their specific movement ways, and according to Angold et al. [38], they require diversity of green spaces. For this reason, it is important to phrase land use policies which provide for the need for biodiversity.

While it is impossible to study exhaustively all species, ecologists focus on some key groups. Birds are among the most studied groups in urban ecology [34,39], and avian species are known as excellent bio-indicators [40]. Moreover, the presence of birds in cities contributes to the soundscapes of cities, investigated in some studies [41–43]. The cited literature show that natural sounds are preferred by city inhabitants. The way the avian biodiversity contributes to the soundscape of the city will be subject to a future study, similar to [44], conducted in Europe by a network on the topic of COST TUD Action TD-0804 Soundscapes of European Cities and Landscapes. Moreover, bird species richness is highly variable across the urban-to-rural gradient [45]. The characteristics of rural individuals are different compared to their urban peers, both at species and population levels [46]. In this context, urban agriculture [47] deserves a special attention.

Ornithological studies investigating urban environments are scarce in North Africa, especially in Algeria, and even more precisely in the north of Algeria. Information is still incomplete, fragmentary, or even lacking, especially for some urban-adapted and urban-exploiter species [48,49]. In the area of Annaba, despite its ecological importance and localization (detailed in the Materials and Methods section), there are too few studies dealing with the avian diversity. The most noteworthy urban studies carried out in the area focused only on single species or groups, considering the phenotype of the Collared Dove (*Streptopelia decaocto*) [50], focusing on the health status of the Feral Pigeon (*Columba livia*) [51] or on the *Columbidae* [52,53].

The aim of this study is to investigate the controversial relationship between biodiversity and urbanization by analyzing the avian diversity in Annaba, Algeria along an

urbanization gradient, using a novel geo-statistical approach. We hypothesize that diversity decreases towards the urban environment, along with the reduction of vegetated areas. Unlike the previous ones, this study investigates all avian species encountered during the field sampling.

## 2. Materials and Methods

This study relies on a novel geo-statistical approach, combining geospatial techniques with statistics. In a nutshell, field observations were mapped, combined with remote sensing data on land cover and use, and spatial analyses carried out based on the statistical processing of data.

### 2.1. Case Study: Annaba, Algeria

The study was carried out in and around the City of Annaba from the Wilaya of Annaba in Algeria. Figure 1 shows the study location in an international and national context, and also the location of sampling points.

Annaba city is the fourth largest city after Algiers, Oran, and Constantine. It is located between the latitude of $7°42'$ and $7°48'$ east, and the longitude of $36°50'$ and $37°57'$ north in the north east of Algeria at 100 km from the Algerian–Tunisian border in the east. The city is at 600 km from the capital Algiers, covers 1412 km$^2$, and its climate is typically Mediterranean, with an average annual temperature of 18 °C and an annual rainfall ranging from 650 to 1000 mm with a peak in winter and deficit in summer [54]. Annaba City is bordered in the north and west by the Massif of Edough, the Mediterranean Sea to the east and the alluvial plain of the Seybouse Wadi to the south. The average population in 2020 is estimated to 715,370 inhabitants throughout the region, including over 358,000 in the city of Annaba [55].

Annaba is one of the oldest cities in Algeria, founded in 1295 BC. The city was named successively Ubon, Hippo Regius, Hippone, Bouna, Bled El Aneb, Bône, and finally Annaba [56]. The antique city was place to the passage of several civilizations: Phoenicians, Punic, Numidian, Roman, and Byzantine. Nevertheless, the Roman remains represent the Roman phase that we see today: cultural buildings, villas, theaters, fountains, etc. [57]. The Arab-Muslim period (7th and 8th century) left as mark a part of the historic downtown of Annaba "Place d´arme" represented by capitols of the composite gallery, porticoes with triple section, wall niches, etc. [58]. In the French Colonial period, the neoclassical style made its appearance [47,59] to the point where the architectural and urban production in Algeria merges with that of France. Until contemporary times, from the 1970s, there was a production of collective buildings under the framework of a more collective architecture characterized by a modern style poor in decoration. This is found especially in the main hotels of the city, along with a post-modern style used especially for shopping centers, and dominated by arches and colors, as well as glass buildings (clinics, banks, administrations, etc.) [60].

Over time, the city of Annaba has experienced increased sprawl, green spaces making way to buildings. The city is mainly made up of buildings with very low vegetation in the surrounding area, and the current trend is to suppress vegetation in order to build homes, commercial places, etc., mainly for economic reasons [61]. There are several predominantly wild forest areas in the periphery, such as those around "La basilique de Saint Augustin", the "19 May 1945" Stadium), the Sidi-Achour road, and especially the mountainous region of "Séraïdi". However, there are fully open parks in the downtown: Cours de la revolution, Placette Alexis Lambert, and others delimited: Boukhtouta Hocine Garden, The Squares of Edough, as well as cemeteries, etc. The most important site from an ecological viewpoint is the Christian Cemetery, which is a closed space inaccessible to the vast majority of population with an area of 6.5 ha, and the largest park in an urban environment, characterized by a substantial avian diversity [62].

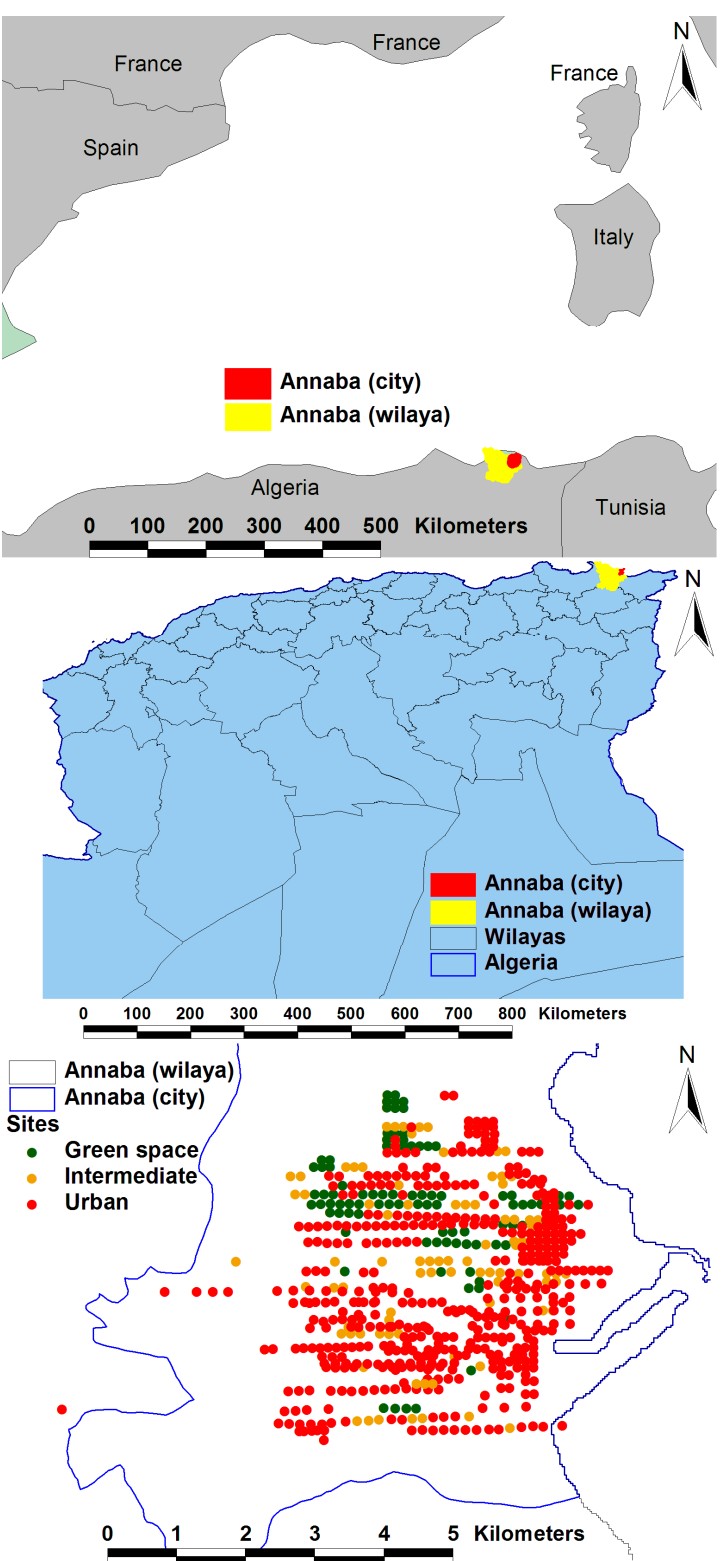

**Figure 1.** Location of the case study in an international (**top**) and national context (**middle**), and of the sampling sites (**bottom**).

## 2.2. Data

The data used in this research comes from two sources: field observations and remote sensing data.

Field observations were acquired during the period of 2017–2018 from 650 sites, classified, following the methodology of Aouissi et al. [63,64], based on the land cover and use data, as "green space"(characterized by the dominance of vegetation, including parks, cemeteries, and wooded green spaces, 92 sites), "intermediate" (sites where the presence of trees and greenery was noted, but with adjacent built areas, 92 sites), and "urban" (where the built up area is predominant, and without green spaces, 466 sites). There were 4245 raw observations, each line including the name of species, number of individuals, coordinates of site, date of observation, type of site, presence of vegetation (%), and whether the observation was made during the reproductive season or not.

We used the method of IPA ("indice ponctuel d'abondance"/point abundance index) [65]. It consists, for an observer, of staying motionless for 15 minutes and recording all individual birds heard or seen [66]. The sampling was performed twice by a single observer (H.A.A) on all 650 sites, previously identified cartographically using GPS. The first sampling was carried out at the beginning of spring and during summer (between 7 February and 1 September), in order to catch the presence of sedentary and breeding migratory species during the breeding season. The second sampling was carried out later on in the season (between 5 September and 1 February), outside of the breeding season. Data was collected on each site twice a day: early in the morning (approximately between 5:00 and 7:00 GMT + 1 in summer or between 6:00 and 8:00 GMT + 1 in winter) and before sunset (approximately between 17:00 and 19:00 GMT + 1 in winter or 18:00 to 20:00 GMT + 1 in summer). In order to avoid overlapping, a minimum distance of 100 m was maintained between any two sites.

Spatial data was derived from satellite imagery, overlaying the distribution of sites with the image and determining the land cover and use for each site according to the classification presented above. The satellite imagery and geographical data, including the administrative limits, was obtained from the Cartography Department of the National Hydraulic Basins Agency (ABH CSM) "Agence de Bassin Hydrographique Constantinois-Seybousse-Mellegue" [67] by M.T.

### 2.3. Data Processing and Analysis

Data was analyzed using the General Linear Models to find the influence of different factors, including year and month of collection, presence of vegetation, and whether the observation was made during the reproductive season or not, on the number of individuals from each species.

Following the statistical analysis, it was found that since only 323 of the total 4245 observations were from 2018, the year did not significantly influence the avian diversity, and the model could not run. As a result, during the next step data was aggregated at the site level, combining all observations for the same species from both years. The resulting data set had 2738 observations, each line containing the site, species, and number of individuals. This data set was used to compute for each site separately two indices of diversity: the species richness, defined as the total number of species per site, and Shannon-Wiener's informational entropy index, computed using Equation (1):

$$H = \sum_{n}^{i=1}(-1) \times p_i \times \ln(p_i) \tag{1}$$

where $n$ is the total number of species, and $p_i$ is the share of individuals from species $i$ from the total number of individuals per site.

The choice of the two indices was motivated by their ability to compare the diversity of different sites [68] or of the same site across time [69] and visualize the distribution in a spatial perspective, when used in conjunction with geo-statistical approaches [70].

The results were analyzed statistically, in order to determine the relationship between land cover and use and biodiversity, using the analysis of variance (ANOVA) followed by three post-hoc comparison tests: Tukey, Bonferroni, and Scheffé. The statistics were computed using Microsoft Excel 2003 (Microsoft, Spring Valley, CA, USA).

The geospatial analyses examined the spatial distribution of biodiversity, assessed using the species richness and Shannon-Wiener's informational entropy index, against land cover and use. For this purpose, ArcGIS10.X (Redlands, CA, USA) was used for the spatial interpolation of the two biodiversity values via simple kriging [71], preferred due to its intuitiveness and broad use. The land cover surfaces were also derived by interpolation based on assigning proximities with the Spatial Analyst extension of ArcView GIS 3.X (Redlands, CA, USA).

## 3. Results

Overall, all results are attempting to answer the question whether avian biodiversity depends on land cover and use, testing its variation across a gradient of urbanization.

### 3.1. Overall View of the Sample

The analysis of the 4245 observations revealed the presence of 28 bird species. Table 1 presents the species identified, showing the total number of individuals, its average numerical distribution per site, and number of sites where the species was found.

**Table 1.** Species found in Annaba, Algeria during 2017–2018. The table shows the total and average number per site and the number of sites where each species was present.

| Species | Total No. | Average No. per Site | No. of Sites Present |
|---|---|---|---|
| *Apus apus* | 108 | 3.72 | 29 |
| *Bubulcus ibis* | 10 | 1.11 | 9 |
| *Chloris chloris* | 117 | 1.65 | 71 |
| *Chroicocephalus ridibundus* | 167 | 4.64 | 36 |
| *Columba livia* | 4424 | 8.27 | 535 |
| *Columba oenas* | 1 | 1.00 | 1 |
| *Columba palumbus* | 1 | 1.00 | 1 |
| *Cyanistes caeruleus* | 86 | 1.62 | 53 |
| *Cyanistes teneriffae* | 67 | 2.23 | 30 |
| *Delichon urbicum* | 66 | 2.20 | 30 |
| *Erithacus rubecula* | 50 | 1.19 | 42 |
| *Falco tinnunculus* | 62 | 1.29 | 48 |
| *Fringilla coelebs* | 161 | 1.96 | 82 |
| *Goeland leucopée* | 102 | 3.09 | 33 |
| *Hirundo rustica* | 329 | 7.65 | 43 |
| *Muscicapa striata* | 112 | 1.33 | 84 |
| *Parus major* | 84 | 1.50 | 56 |
| *Passer domesticus* | 1763 | 5.23 | 337 |
| *Phoenicurus ochruros* | 113 | 1.36 | 83 |
| *Phylloscopus collybita* | 132 | 1.94 | 68 |
| *Pycnonotus barbatus* | 116 | 1.38 | 84 |
| *Serinus serinus* | 114 | 1.90 | 60 |
| *Spilopelia senegalensis* | 117 | 2.72 | 43 |
| *Streptopelia decaocto* | 5745 | 9.28 | 619 |
| *Streptopelia turtur* | 74 | 3.36 | 22 |
| *Sturnus vulgaris* | 158 | 6.32 | 25 |
| *Sylvia atricapilla* | 189 | 2.08 | 91 |
| *Turdus merula* | 517 | 4.17 | 124 |

The view of the entire sample indicates a diverse presence, with clear outliers: the Feral Pigeon (*Columba livia*) and the Collared Dove (*Streptopelia decaocto*) were the most numerous, and also present in most sites in large numbers. The Stock Dove (*Columba oenas*) and the Common Wood Pigeon (*Columba palumbus*) were at the opposite extreme, each found only once in a single site (note that these species were completely absent from our surveys during previous counts). The number of individuals per site also depends on the

behavior of species, as some of them tend to group (e.g., *Sturnus vulgaris*, *Passer domesticus*), and others to spread out (for example, the predator *Falco tinunculus*).

The results of statistical analysis of the overall data set are presented in Table 2. The table combines two General Linear Models; the first one, labeled "full model", includes all variables, and the second, labeled "prediction model", only those statistically significant. The first model was overall statistically significant, with F = 12.83 and $p < 0.0001$; it had an adjusted $R^2$ of 0.043524. The second model was overall statistically significant, with F = 13.71 and $p < 0.0001$; it had an adjusted $R^2$ of 0.043419. The table indicates that the month of observation, land cover and use, and presence of vegetation have a statistically significant influence on the number of individuals from each species.

**Table 2.** General Linear Models examining the relationship between the number of individuals from each bird species found in Annaba, Algeria during 2017–2018 and the variables with a potential influence on it.

| Variable | Full Model | | | | | Prediction Model | | | | |
|---|---|---|---|---|---|---|---|---|---|---|
| | DF | Type III SS | Mean Square | F Value | Pr > F | DF | Type III SS | Mean Square | F Value | Pr > F |
| Month | 11 | 1021.83 | 92.89 | 8.60 | <0.0001 | 11 | 1175.08 | 106.83 | 9.90 | <0.0001 |
| Land cover and use | 2 | 724.48 | 362.24 | 33.55 | <0.0001 | 2 | 727.05 | 363.52 | 33.68 | <0.0001 |
| Presence of vegetation | 1 | 103.29 | 103.29 | 9.57 | 0.0020 | 1 | 102.86 | 102.86 | 9.53 | 0.0020 |
| Reproductive season | 1 | 5.02 | 5.02 | 0.47 | 0.4953 | — | — | — | — | — |

### 3.2. Relationship Between Biodiversity and Land Cover and Use

The results of the analysis of variance (ANOVA) indicate that there are statistically significant differences between the three types of sites ("green space", "intermediate", and "urban") with respect to species richness (F = 440.77, $p < 0.0001$) and Shannon-Wiener's informational entropy index (F = 177.17, $p < 0.0001$). The results were confirmed by all three post-hoc tests, which found significant differences between the values of the two indicators among each possible pairs of sites. Figure 2 presents the average values of the two indices, indicating that the underlying hypothesis of the study is verified by the statistical analysis.

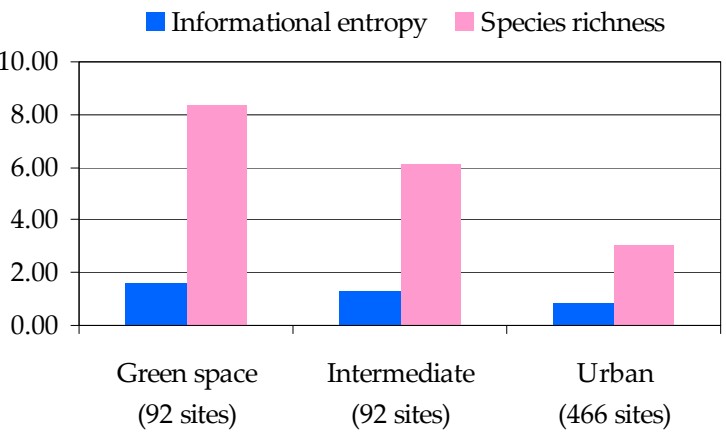

**Figure 2.** Distribution of Shannon-Wiener's informational entropy index and species richness across the gradient of urbanization in Annaba, Algeria. The image shows that biodiversity tends to increase when moving from the built-up area towards the natural habitats.

The same results were obtained from geospatial analysis. Figure 3 presents the spatial distribution of Shannon-Wiener's informational entropy index and species richness, interpolated from the sampling sites; it can be seen that green spaces and intermediate areas are located in areas with higher diversity.

In Figure 4, the same distribution is overlaid with the land cover and use derived from the spatial interpolation of sampling sites. The image shows that areas with high biodiversity (dark shades) overlap with green spaces and intermediate areas.

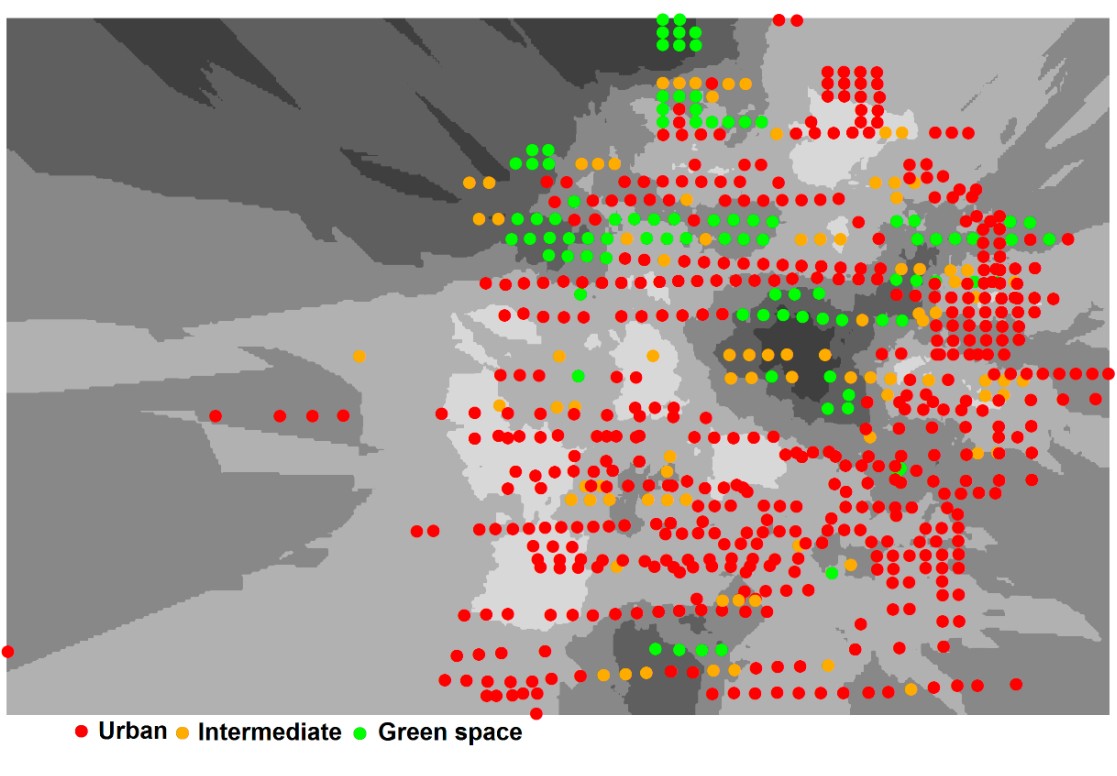

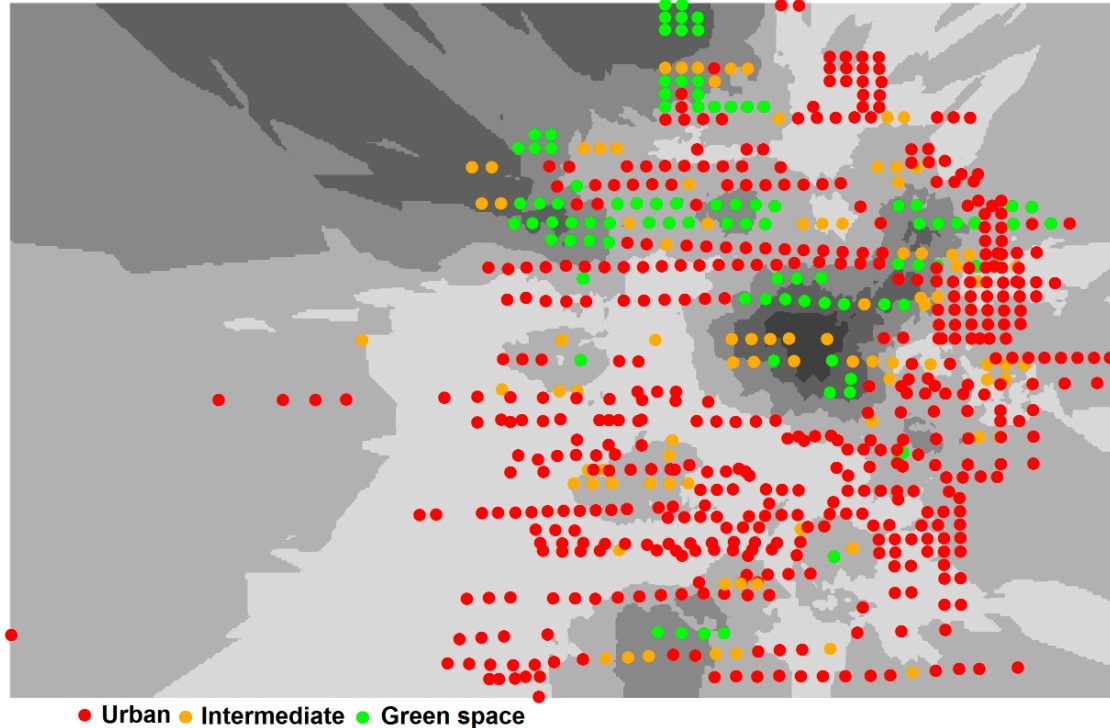

**Figure 3.** Interpolated spatial distribution of Shannon-Wiener's informational entropy index (**top**) and species richness (**bottom**) in Annaba, Algeria. The darker shades correspond to higher values of the two indices, indicating higher biodiversity.

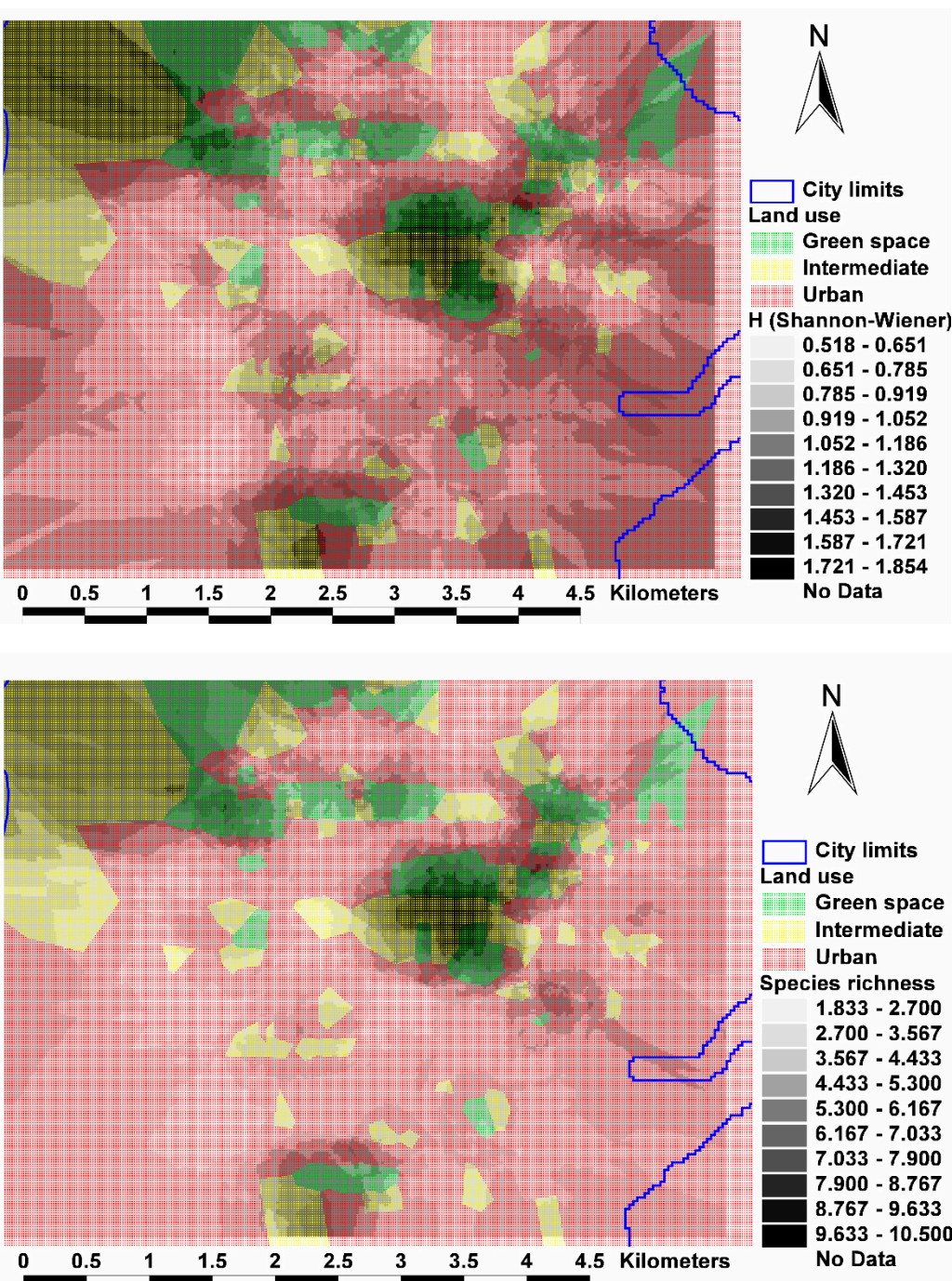

**Figure 4.** Interpolated spatial distribution of Shannon-Wiener's informational entropy index (**top**) and species richness (**bottom**) and interpolated land cover and use in Annaba, Algeria. The darker shades correspond to higher values of the two indices, indicating higher biodiversity.

## 4. Discussion

### 4.1. Significance of the Findings and Comparison with Similar Studies

This study aimed to test the hypothesis according to which biodiversity is inversely proportional with the degree of urbanization. The statistical analyses confirmed the hypothesis, showing than the biodiversity of green spaces is significantly greater than the one of intermediate sites, which in its turn is significantly greater than the one of urban sites. These results bring additional evidence to the fact that biodiversity is negatively influenced by urbanization [30,31], leaning the balance of the existing controversies in this direction. At the same time, spatial analyses cast a new light on the potential implications, especially

in relationship to planning, discussed in the next section. Last but not least, the results have a special importance from a methodological perspective, indicating the importance of combining statistical approaches with geo-spatial techniques in order to get a detailed picture of the spatial distribution of biodiversity.

Several similar studies were carried out in other regions. In a study carried out in Mar del Plata City, southeastern Buenos Aires Province, Argentina [72], 39 species were counted; the most abundant was the Feral Pigeon, and there was no relationship between the level of urbanization and stability of bird species richness. Another study considered three Swiss cities: Zurich, Lucerne, and Lugano [73]; in 96 sampling points, 63 species were found, generally outside of the cities, while in highly urbanized areas, only few were found, namely *Columba livia* and *Passer domesticus*, which is very similar to our results. The study showed that bird species richness and diversity were negatively affected by increasing sealed area or buildings, and positively influenced by increasing vegetation structures, in particular trees. A bird count was carried out in Algeria using the same method in a much less urbanized area (Ziban region); 42 species were found, most of them from the Turdidae family [74].

The relationship between avian biodiversity and urbanization in the Mediterranean climate has been addressed by Di Pietro et al. [75] and Vignoli et al. [76], both investigating it in the metropolitan area of Rome. The first study, focusing on breeding birds, showed that the effects of urbanization are little investigated in other species-area relationship studies in urban ecology. The second study, based on 69 breeding bird species, compared the urban–rural gradient between the core of metropolis and surrounding area, subdividing the city in cells classified according to landscape, and the highest heterogeneity was found for the rural gradient. The authors explicitly called for studies in other cities in order to validate their findings. We consider these studies relevant for comparison due to being carried out in the same climate region as in Algeria and taking into account the reference to the green planting map of Rome. The 2013 Le Notre Landscape Forum in Rome also dealt with the urban periphery, which is worthy considering in the context of urban sprawl [77]. Vignoli et al. [76] looked specifically at this periphery as a heritage of urban villas in a city inhabited for millennia. The decrease of biodiversity was observed at point count and landscape scale. The study showed that different species exhibit different trends in the rural-urban gradient. Although the agricultural land was richer in species, this is also true for urban villas, now part of the city, which are remnants of the former countryside and have large gardens. This sustains our findings regarding the relationship between the green spaces and avian biodiversity, even if the green spaces are disconnected. The conclusion of Vignoli et al. [76] is that green areas in the city (such as the gardens of the villas) provide avian biodiversity to the inner-city areas. In Rome, they are replacing parks (for example the villa Borghese garden), with implications calling for the proper conservation of these areas.

Our study is one of only few such studies that have been conducted in Northern Africa. Provided that Algeria in general remains very little studied, and the study region specifically, this study fills in an important gap in the knowledge of North African avian biodiversity, urban environment in general, and the biodiversity related to this type of habitats is often overlooked. In the majority of cases, studies were carried out only outside of urbanized areas until recently, when an interest in the city started developing, as evidenced by the study of Belabed-Zediri [78], focused on water birds from urban areas. The idea that cities are regarded as deserts from an ornithological point of view tends to be abandoned over time [72,79].

In our study, even if the results show a decrease in bird diversity along the urbanization gradient, they also show an important richness of green spaces within the urban areas, and the presence of same species found outside the city. A high concentration of diversity and abundance in the environments with significant vegetation should be noted, testifying to the importance of green spaces in urban environments, further discussed in the next section in relationship to planning.

### 4.2. The Planning Perspective

The planning consequences rely on previous studies. Planners are considered responsible for developing a strategy for urban green infrastructure [30,68,80]; even taking it into account in planning and management can increase urban sustainability [81–87], by turning the vicious circle determined by its fragmentation into a virtuous one [88] when creating corridors to link natural patches. The strategy must be built on the premise that conservation should protect networks instead of parcels [30]; this is vital to maintaining ecosystem services [26]. The possible solutions include restoring connectivity [89] or buffering natural areas [32], and linking protected areas or parks [17]. Most important, the strict enforcement of planning provision makes an important contribution to the conservation of biodiversity, in opposition to any deviation from it by the means of derogatory planning [12,90].

In the particular case of this study, the lesson according to which biodiversity decreases along the urbanization gradient implies a need for solutions aimed at connecting the biodiversity hotspots, i.e., green spaces. Provided that the "intermediate" sites also show a high biodiversity, significantly higher than the urban ones, they can be used as links between the green spaces, ensuring the movement of species across sites.

In Northern Europe a decline was observed in the sparrow population, despite many environmental restoration activities (e.g., unsealing the ground through permeable plates in parking places etc.) which is highlighted by the literature [91]. This underlines that the urbanization influence on the decline of avian biodiversity is not reversible if the area undergoes ecological restoration afterwards [73], and thus the efforts to preserve the initial ecological status have to be done. This was also present in the COP21 prescriptions, and caused citizen participation movements in Northern Europe against transport infrastructure extension around cities, which can lead to the loss of biodiversity in cities and the agricultural land around them [92]. The conflicts in urban management between stakeholders are not to be decoupled from the energy discussion by COP21, but include the debate on the neighborhoods at the city periphery with single family housing. The Algerian urban model follows, as stated earlier, the French Western European urban model with urban sprawl occurring in the periphery through the emergence of new neighborhoods in the 20th century. Instead, the "house with garden" model, also of French influence (see also [93]), should be preserved to provide not only for connections, but also for the diversity of green spaces. Currently low-rise housing and green spaces are endangered by speculative building. Preserving the green areas of the city and recycling the historic building stock also helps moving to a circular economy and countering the urban heat islands, with a positive effect on vegetation and biodiversity.

Few other studies considered the relationship with planning practices. In Canoas, the metropolitan area of Porto Alegre (Brazil), 100 species were found, mostly doves; their number was negatively affected by the increase of urbanization (and noise level), but the share of green areas had a positive effect. Nevertheless, the study found that the presence of particular habitats (wetlands, grasslands, woodlots), patchily distributed in the urban matrix, could buffer the effects of urbanization on birds, and recommended taking them into account in urban planning. The study recommended supporting citizens in maintaining residential vegetation (e.g., private yards), and, hence, keeping native vegetation areas inside the urban area, to increase the green areas and promote biological conservation [94]. In Valdivia (Chile), green spaces had a positive impact on the distribution and diversity of 32 bird species; various categories of green space can have very different effects even exerting a negative influence, such as the municipal green areas. Municipal green spaces, designed and maintained for recreational purposes, were found more homogeneous with respect to land cover and vertical heterogeneity when compared to non-municipal green areas. The authors recommended planners pay a special attention to municipal green areas, where habitat quality for birds can be improved by reducing the impervious surface and creating and conserving a multi-layered vegetation structure, and preserve non-municipal green areas, including wetlands critically threatened by urban development [95]. In the municipalities of Vancouver and Burnaby (British Columbia), 25 common species were

found; the results suggested that both local and landscape-scale resources were important in determining the distribution of birds in urban areas. They recommended the integration of parks, reserves, and surrounding residential areas into urban planning to preserve the diversity of resident avifauna and overall species diversity. Also, development on the verge of continuously forested areas was found useful to minimize the impervious surface cover and house size, maintain native tree and berry shrubs, integrate new ponds, and preserve and develop natural freshwater sources. Residential areas near parks were found to be more likely to recruit sensitive nesting species and probably experience frequent use by species from nearby parks [19].

### 4.3. Future Research Directions

Our study could be extended to explore biodiversity indicators for many dimensions; we could, for example, verify the stability of the explicative model of biodiversity during our period (2017–2018), previously, and for the future periods using several statistical tests such as the Chow Test, etc. However, currently it is difficult to do investigative work, in particular because of the epidemiological crisis due to the COVID-19 pandemic, especially in Algeria [96].

Future research could also explore more explicative factors of biodiversity, such as industry, agricultural activity, demographic growth, ecological indicators, etc. Urban agriculture can be a determining factor. In Algeria [97], but also in the European Union [98], attention is given, apart of green spaces, to animal species in urban agriculture. Therefore, not only recreation areas, but also production areas can connect the green areas of the city with the peri-urban ones.

It would be interesting to extend the study area to several cities, and make comparisons, for example, between semi-arid and arid areas of Algeria; this may help clarifying better the relationship between urbanization and bird communities. At least, especially in North African and Middle-East regions, this kind of studies remain scarce and all additional data can be considered important. A comparison can be made between areas around the Mediterranean, including the European ones.

The debate can also be carried towards the role of individual dwellings versus collective housing. In each case, adequate green public space has to be assured. In cities which do not have historical fortifications, individual homes with gardens proliferate. On the other hand, the spread of the International Style favored the construction of blocks of flats in the green areas. These green spots, rarely mapped, contribute to linking green spaces. A representative project was carried out in Rome, Italy to map green spaces [99]. A similar study can be developed based on an urban analysis.

### 5. Conclusions

This study has explored the hypothesis according to which biodiversity, and in particular avian diversity, decreases from the urban towards the natural areas. The hypothesis was confirmed by the findings, providing some important insight from a planning perspective. First, urban green spaces are important biodiversity hotspots, and second, connectivity is important to link them and permit the migration of birds from one area to another. From a planning perspective, this study brings additional evidence for the need for connecting the green infrastructure, enabling corridors for the avian fauna. At the same time, the research is relevant for its region, as the North-African biodiversity is insufficiently explored in the literature, and also for the coastal Mediterranean environments. Future studies could make use of the data and results and check whether the patterns are comparable with those from Europe and elsewhere. Last but not least, the study underlines the potential of diversity indices, such as Shannon-Wiener or species richness, for being used in a spatial setting, with outcomes relevant both for planning and scientific purposes.

**Author Contributions:** Conceptualization, H.A.A., A.-I.P. and M.B.-D.; methodology, H.A.A. and A.-I.P.; software, A.-I.P. and M.T.; validation, A.-I.P., M.B.-D. and Z.B.; formal analysis, A.-I.P. and M.A.; investigation, H.A.A.; resources, H.A.A. and M.T.; data curation, A.-I.P. and M.A.; writing—

original draft preparation, H.A.A., A.-I.P. and M.B.-D.; writing—review and editing, H.A.A., A.-I.P. and M.B.-D.; visualization, H.A.A., A.-I.P. and M.B.-D.; supervision, A.-I.P., M.B.-D. and Z.B. All authors have read and agreed to the published version of the manuscript.

**Funding:** This research was funded by the DGRSDT and MESRS (Algerian Ministry).

**Data Availability Statement:** The data presented in this study are collected within a research project and can be made available on request from the corresponding author.

**Acknowledgments:** Many thanks are addressed to Samir Merouana for his assistance in data entry, Anis Aouissi and Aissam Gaagai for their help with GIS, and the DGRSDT and MESRS for their support.

**Conflicts of Interest:** The authors declare no conflict of interest. The funders had no role in the design of the study; in the collection, analyses, or interpretation of data; in the writing of the manuscript; or in the decision to publish the results.

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
