# Peer review of "Influence of Land Use on Avian Diversity in North African Urban Environments"

_land, doi:10.3390/land10040434_

Round 1

Reviewer 1 Report

Introduction section needs an improvement, please provide more literature reviews.

In figure 1, sign of north should not be inside map. It is better if you shifted to corner of map.

Figure 2 needs more details and more descriptions.

Explain more about Figure 4.

Figure 5 requires more description.

Compare result finding by other researchers’’ results.

Provide some recommendation in Conclusions section.

There are too much self-citations. This is not allowed. Please fix it accordingly.

Reviewer 2 Report

Aouissi et al's study examined relationships between urbanization and avian biodiversity in Annaba, Algeria. The authors determined that avian richness was greatest in areas with less urban development, and least in areas of intense urbanization. They discuss the implications their findings should have for future urban planning practices. 

I liked this study and believe similar studies like this one should be conducted globally to see if this relationship between species richness and urbanization holds across continents (I suspect it does). However, this manuscript would benefit from a few clarifications/modifications:

Overall: There are numerous typos throughout the manuscript and should be edited carefully before the next submission. Further, some phrasing  translates awkwardly into English and those should be modified/removed. 

Abstract: Check for typos here and there. I would add something at the end of the abstract promoting the idea that this study could help improve urban planning policy as so much of the discussion details this possibility.

Introduction: The introduction is very long. I would recommend removing paragraph 4 in its entirety and a portion of paragraph 5 (Lines 83-88). 

Methods: This section is overly long in some portions, but lacks detail in others. I recommend removing paragraph 3 (lines 128-142) as this doesn't really help the reader understand the study. Paragraph 4 (lines 143-154) also gives a lot of information, but it is unclear if the sites detailed here were actually surveyed in the study. If these areas had sampling points within them, please state which ones and how many points were in each. If not, then this paragraph can be removed from the section. 

My biggest issue with this manuscript is the lack of detail concerning how the point abundance index was performed at sampling sites. More detail should be given, even if a reference is provided. A brief description should be added that includes: 1) How many months out of each year had areas surveyed?) 2) Were all areas sampled evenly across the year (e.g. Were green, intermediate, and urban areas ALL sampled in winter months, spring months, etc.; How many were sampled/month?); 3) How long did the sampling periods last? 10 minutes? 1-hour? Was this consistent across all sites?; 4) What time of day did sampling occur? Morning between 0600-1000h? Afternoon hours? Was this sampling time consistent across sites/months of the year? 

I ask for this clarification because the activity patterns of birds can vary by species, sex, season, and reproductive stage. Consistency is important in point sampling to make the different sites truly comparable. 

Results: Line 183 states that year 2018 findings did not have significant influence. Were data from years 2017 and 2018 therefore combined in a new model? Or was 2018 excluded? This point would be easily clarified. 

The description for the Shannon Weiner index is a bit awkward. Perhaps consider something like "n is the total number of species, and pi is the proportion of the total sample represented by species i per site". 

Figure 2 is very busy and challenging to read as so much information is being presented at once. I believe Table 1 is a more concise way of sharing all of this information. However, Figure 2 could be split into 3 separate figures to convey the same messages, but in a more digestible way for the reader. 

Discussion: This section mostly emphasizes the importance of urban planning practices, which is a great application of the findings of this study. However, the reader might appreciate an additional reference or two concerning specific avian species in which other similar studies have attempted to enhance their conservation efforts through better planning practices.  The sparrow population in Northern Europe was mentioned. Are there any others? 

Round 2

Reviewer 1 Report

The authors improved manuscript accordingly, and this manuscript is acceptable.

Author Response

Thank you very much for the comments. We are glad that we were able to address all these constructive remarks, that helped us improving the quality of the manuscript.